# Efficient DNA-free genome editing of bread wheat using CRISPR/Cas9 ribonucleoprotein complexes

Zhen Liang[1,2,*], Kunling Chen[1,*], Tingdong Li[1,2], Yi Zhang[1,2], Yanpeng Wang[1,2], Qian Zhao[1,2], Jinxing Liu[1], Huawei Zhang[1], Cuimin Liu[1], Yidong Ran[3] & Caixia Gao[1]

Substantial efforts are being made to optimize the CRISPR/Cas9 system for precision crop breeding. The avoidance of transgene integration and reduction of off-target mutations are the most important targets for optimization. Here, we describe an efficient genome editing method for bread wheat using CRISPR/Cas9 ribonucleoproteins (RNPs). Starting from RNP preparation, the whole protocol takes only seven to nine weeks, with four to five independent mutants produced from 100 immature wheat embryos. Deep sequencing reveals that the chance of off-target mutations in wheat cells is much lower in RNP mediated genome editing than in editing with CRISPR/Cas9 DNA. Consistent with this finding, no off-target mutations are detected in the mutant plants. Because no foreign DNA is used in CRISPR/Cas9 RNP mediated genome editing, the mutants obtained are completely transgene free. This method may be widely applicable for producing genome edited crop plants and has a good prospect of being commercialized.

[1] State Key Laboratory of Plant Cell and Chromosome Engineering, Institute of Genetics and Developmental Biology, Chinese Academy of Sciences, Beijing 100101, China. [2] University of Chinese Academy of Sciences, Beijing 100049, China. [3] Genovo Biotechnology Co., Ltd., Tianjin 301700, China. * These authors contributed equally to this work. Correspondence and requests for materials should be addressed to C.G. (email: cxgao@genetics.ac.cn).

The CRISPR/Cas9 system has been widely used in plants to introduce genome modifications, and is paving the way for precision crop trait improvement[1]. Normally, CRISPR/Cas9 DNA constructs are delivered into plant cells by *Agrobacterium tumefaciens* mediated T-DNA transfer or biolistic bombardment, become expressed, cleave target sites and produce mutations[2]. During this process, there is a strong possibility that the CRISPR/Cas9 constructs are integrated into the plant genome[3]. This increases the chance of producing unwanted genetic changes, the most important of which are transgene integration and off-target mutation. Furthermore, once inside the recipient cells, the CRISPR/Cas9 sequence may be degraded, and the resulting fragments can serve as filler DNA in the double stranded break repair process and become inserted into intended and/or unintended genomic sites[4–6]. Thus, at present, the biosecurity of genome-edited plants is an important public concern[7].

In response to this concern, substantial efforts are being made to optimize CRISPR/Cas9 mediated genome editing with the aim of avoiding transgene integration and off-target mutations. Recently, we showed that transient expression of CRISPR/Cas9 DNA or RNA (TECCDNA or TECCRNA) in wheat resulted in efficient genome editing with significantly reduced transgene integration[3]. Moreover, Woo *et al.*[8] demonstrated that the use of preassembled CRISPR/Cas9 ribonucleoproteins (RNPs) completely avoided transgene integration, and greatly decreased off-target mutations. They delivered the RNPs into lettuce protoplasts and obtained transgene-free mutant plants. But plant regeneration from protoplasts is technically challenging in many species, especially in monocot crops. Therefore, we aimed to develop a CRISPR/Cas9 ribonucleoprotein mediated genome editing method for efficient and specific genome editing of major monocot crops using hexaploid bread wheat (*Triticum aestivum* L., AABBDD, $2n = 6x = 42$) as experimental material.

## Results

**Validation of RNPs in wheat protoplasts.** As the first step in our work, we tested if CRISPR/Cas9 RNPs may cleave targeted genomic sites and induce mutations in wheat protoplasts using the gw2-sgRNA that we previously found to be highly active on the *TaGW2* gene[3]. This gene functions in grain weight control and has three very similar homoeologs (*TaGW2-A1, -B1* and *-D1*) in bread wheat[9]. The guide RNA sequence of gw2-sgRNA, targeting a conserved region in exon 8, matched perfectly its recognition site in *TaGW2-B1* and *-D1* (Fig. 1a; Supplementary Table 1). However, there was a single nucleotide mismatch at the cognate target site in *TaGW2-A1*, which was located at position 9 of the PAM-proximal region of the sgRNA (Fig. 1a). We expressed Cas9 protein from *Escherichia coli*, purified it and complexed it with *in vitro* transcribed gw2-sgRNA. The resulting RNPs (gw2-RNPs hereafter) displayed robust cleavage activity *in vitro* (Supplementary Fig. 1a). Subsequently, the gw2-RNPs were transfected into wheat protoplasts, and potential editing activity was detected by PCR-RE assay[10]. The on-target mutagenesis frequencies for *TaGW2-B1* and *-D1* induced by gw2-RNPs were 33.4% and 21.8%, respectively, whereas the off-target editing frequency for *TaGW2-A1* was 5.7% (Fig. 1b). In a parallel protoplast transfection experiment involving the expression of Cas9 and gw2-sgRNA from a plasmid DNA construct (pGE-TaGW2)[3], the mutagenesis frequencies for *TaGW2-B1* and *-D1* were 41.2% and 35.6%, respectively, whereas that for off-target editing of *TaGW2-A1* was 30.8%. Thus, the on-target mutagenesis frequencies induced by gw2-RNPs were comparable to those induced by pGE-TaGW2, while the off-target editing frequency produced by gw2-RNPs was over five-fold lower than that produced by pGE-TaGW2.

**The effectiveness of gw2-RNPs in immature embryo cells.** Having confirmed that gw2-RNPs were highly active in wheat protoplasts, we delivered them into the immature embryo cells of the bread wheat variety Kenong 199 by particle bombardment (see Methods). In parallel, we carried out TECCDNA genome editing with the plasmid pGE-TaGW2. Two days after the bombardment, we isolated genomic DNA samples from 100 pooled immature embryos that had been treated with either gw2-RNPs or pGE-TaGW2, and mutations in *TaGW2-A1, -B1* and *-D1* were analysed by deep amplicon sequencing, respectively. The PCR amplicons for the three homoeologs were barcoded and sequenced using Illumina with the depth varying from 80,000 to 400,000 reads. The on- and off-target mutagenesis frequencies were calculated as the ratio of reads containing indels in the target site over the total reads captured. We found that the on-target mutagenesis frequencies of *TaGW2-B1* and *-D1* induced by gw2-RNPs (0.18% and 0.21%, respectively) were about five-fold lower than those induced by pGE-TaGW2 (0.99% and 1.00%, respectively) (Fig. 1c; Supplementary Table 2). Remarkably, the off-target mutagenesis frequency of *TaGW2-A1* produced by gw2-RNPs was a mere 0.03%, six to seven times lower than the on-target mutagenesis frequencies of *TaGW2-B1*

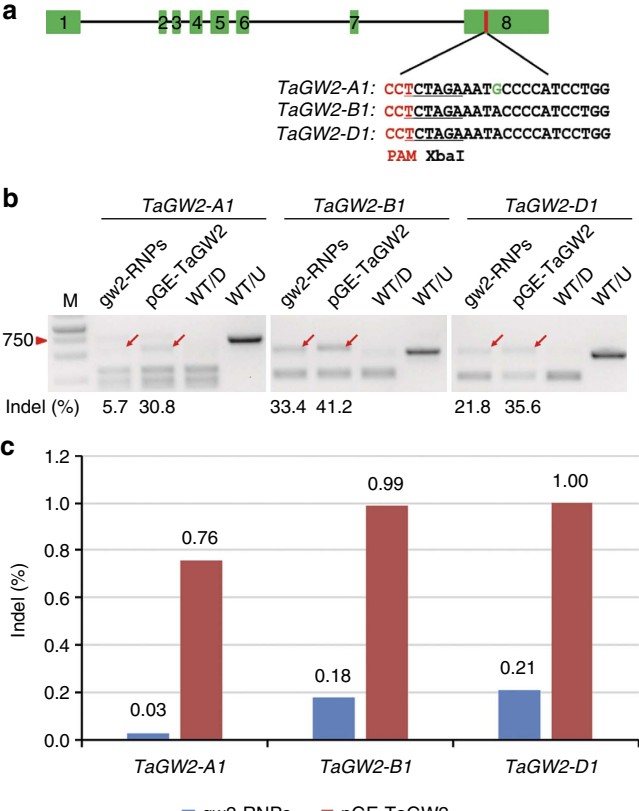

**Figure 1 | Development and validation of CRISPR/Cas9 RNP-mediated genome editing in wheat.** (**a**) The exons of *TaGW2* and the target site of gw2-sgRNA in exon 8. The single nucleotide polymorphism in the targeted sequence of *TaGW2-A1* and the PAM motif are highlighted in green and red, respectively. The XbaI restriction site is underlined. (**b**) Mutagenesis frequencies of *TaGW2-A1, -B1* and *-D1* (induced by gw2-RNPs or pGE-TaGW2) in wheat protoplasts analysed by PCR-RE assay. Mutation bands are indicated by red arrows. WT/D and WT/U indicate wild type PCR amplicons with or without restriction enzyme digestion. (**c**) Mutagenesis frequencies of *TaGW2-A1, -B1* and *-D1* in embryos treated with gw2-RNPs or pGE-TaGW2 revealed by deep amplicon sequencing.

and -D1 (0.18% and 0.21%, respectively) (Fig. 1c). In contrast, with pGE-TaGW2, the off-target mutagenesis frequency of *TaGW2-A1* (0.76%) was only slightly lower than the on-target mutagenesis frequencies of *TaGW2-B1* (0.99%) and -D1 (1.00%). The most frequently occurred mutation types are listed in Supplementary Fig. 3. These results show that gw2-RNPs successfully enter wheat cells and induce indels at the target sites. gw2-RNPs also resulted in a much higher ratio of on-target to off-target editing events than pGE-TaGW2.

**Analysis of T0 mutants.** To produce a population of wheat mutants for analysing the effects of gw2-RNPs in detail, bombarded immature embryos were allowed to regenerate as described previously[3]. In the absence of selection in the medium, plantlets were readily regenerated in six to eight weeks. They were then screened for the presence of targeted mutations in *TaGW2* (see Methods). In total, we obtained 28 *tagw2* mutants from 640 immature embryos treated by gw2-RNPs, with the mutant production efficiency (MPE) being 4.4% (28/640). In the parallel TECCDNA editing experiment in which 640 immature embryos were bombarded with pGE-TaGW2, 30 *tagw2* mutants developed, yielding an MPE of 4.7% (30/640). We analysed these mutants by PCR-RE with three primer pairs (Supplementary Table 3) specific for each of the three *TaGW2* homoeologs, followed by mutation validation by Sanger sequencing. Of the 28 mutants induced by gw2-RNPs, 14 contained indels in *TaGW2-B1* and 28 in *TaGW2-D1*, while none had mutations in *TaGW2-A1* (Fig. 2a,b). Thus, among the mutants generated by gw2-RNPs, the mutation frequencies of the three homoeologs (*TaGW2-B1*, -D1 and -A1) were 2.2% (14/640), 4.4% (28/640) and 0% (0/640), respectively (Fig. 2c). Both heterozygous and homozygous mutations were found for *TaGW2-B1* and -D1, with

each mutant carrying 1 to 3 mutated alleles (Fig. 2b,d). A full description of the mutations detected in the 28 mutants is provided in Supplementary Table 4. Among the 30 *tagw2* mutants obtained using the TECCDNA method, the mutation frequencies of *TaGW2-B1* and *TaGW2-D1* were 4.1% (26/640) and 4.4% (28/640), respectively; and consistent with our earlier finding[3], *TaGW2-A1* was also mutated with a frequency of 3.8% (24/640) (Fig. 2c). Moreover, the majority of the 30 mutants contained four or more mutated alleles (Fig. 2d).

**Reducing off-target mutations via the delivery of RNPs.** The occurrence of off-target mutations is a major concerns affecting the use of genome edited plants for basic function analysis and plant molecular breeding. Use of the CRISPR/Cas9 RNP complex, which can be quickly degraded *in vivo*, reduces off-target effect in human cells[5]. *TaGW2-A1*, which contains a one nucleotide mismatch against the GW2-sgRNA can be viewed as an off-target site. Notably, we did not detect any mutations in *TaGW2-A1* among the 28 mutants regenerated from immature embryo treated by gw2-RNPs (Fig. 2b). As a further step in this direction, we checked if off-target mutations might occur at an additional set of 20 genomic sites with two to five nucleotide mismatches with the guide RNA sequence of gw2-sgRNA in the 28 mutants obtained by using gw2-RNPs. No off-target mutations were detected by either PCR-RE assay or Sanger sequencing of specific PCR amplicons (Supplementary Table 5). These results, together with the data obtained in the protoplasts and deep sequencing assay, show that CRISPR/Cas9 RNPs function in a highly specific manner.

**Applicability of the RNP based method in wheat.** The use of CRISPR/Cas9 RNPs for editing other wheat gene and in other wheat backgrounds was investigated. In this case, we used the

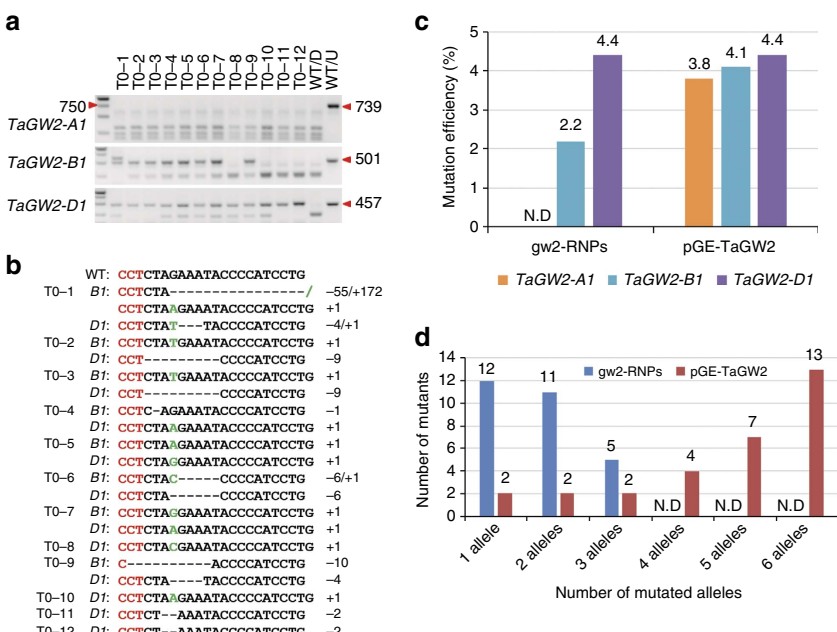

**Figure 2 | Targeted knockout of *TaGW2* by delivery of CRISPR/Cas9 ribonucleoproteins.** (**a**) PCR-RE assay results for 12 representative *tagw2* mutants induced by gw2-RNPs. Lanes T0-1 to T0-12 show the PCR products of the 12 mutants after XbaI digestion. Lanes labelled WT/D and WT/U are the PCR products amplified from wild type (WT) plants with and without XbaI digestion, respectively. The sizes (bp) of the undigested WT amplicons for *TaGW2-A1*, -B1 or -D1 are indicated on the right side. (**b**) Indels caused by gw2-RNPs in the *TaGW2-B1* and/or D1 homoeologs in 12 representative mutants. Hyphens denote deleted nucleotides. Nucleotides inserted are labelled green. The PAM motif (CCT) is shown in red. (**c**) Comparison of on-target (*TaGW2-B1* and *TaGW2-D1*) and off-target (*TaGW2-A1*) mutation efficiencies induced by CRISPR/Cas9 RNPs (with gw2-RNPs) and TECCDNA (with pGE-TaGW2). N.D., not detected. (**d**) Distribution of the individuals carrying 1–6 mutated alleles among the 28 mutants generated by gw2-RNPs and the 30 mutants induced by pGE-TaGW2. N.D., not detected.

*TaGASR7*/gasr7-sgRNA combination because we had previously shown that the gasr7-sgRNA transcribed from plasmid pGE-TaGASR7 could cause indels in *TaGASR7* (Supplementary Table 1)[3]. We found that the RNPs composed of Cas9 and gasr7-sgRNA were highly active in the *in vitro* cleavage assay (Supplementary Fig. 1b). In wheat protoplasts, gasr7-RNPs gave a mutagenesis frequency of 45.3%, close to that induced by pGE-TaGASR7 (48.6%) (Supplementary Fig. 2). Deep amplicon sequencing using genomic DNA extracted from immature embryos of variety Kenong 199 two days after the delivery of gasr7-RNPs revealed a mutagenesis frequency of 0.56% for *TaGASR7* (Supplementary Table 2). The gasr7-RNPs were also delivered into 400 immature embryos of an alternative bread wheat variety (cv YZ814), and we identified seven *tagasr7* mutants by PCR-RE (Supplementary Fig. 4a). Thus, the MPE for this test was 1.8% (7/400). The gw2-RNPs targeting *TaGW2* (see above) were also introduced into YZ814, yielding an MPE of 1.3% (5/400) obtained (Supplementary Fig. 4b).

## Discussion

In this work, we demonstrated that CRISPR/Cas9 RNPs delivered into immature embryo cells of bread wheat by particle bombardment were effective in performing targeted genome editing. We succeeded in editing two different genes (*TaGW2* and *TaGASR7*) in two varietal backgrounds (Kenong 199 and YZ814). In our model variety (Kenong 199), the MPE (4.4%) was relatively high, with four to five independent mutants obtained from 100 bombarded immature embryos. The main steps of CRISPR/Cas9 RNP mediated wheat genome editing include RNP preparation, RNP functional validation, RNP coating and delivery, plantlet regeneration, and mutant identification (Fig. 3). The whole protocol can be performed in seven to nine weeks.

The most important advantage of CRISPR/Cas9 RNP mediated genome editing is the elimination of transgene integration and small DNA insertions in the mutants generated[11]. This is highly desirable for public acceptance of genome edited plants and their use in practical crop improvement. Considering the overwhelming importance of wheat as a major staple food crop worldwide (http://faostat.fao.org/site/339/default.aspx), it is vital to develop an efficient CRISPR/Cas9 RNP based genome editing to promote its genetic improvement. However, as wheat is recalcitrant to regeneration from protoplasts, we have established a procedure for particle bombardment mediated delivery of RNPs into immature embryo cells. This procedure, coupled with our previously optimized methods for selection-free wheat regeneration and high throughput mutant screening[3], has enabled us to edit different wheat genes using CRISPR/Cas9 RNPs. The different steps of our method and the technical details involved in each step may be useful for genome editing studies in other crops using CRISPR/Cas9 RNPs. This suggestion may be particularly relevant for the major cereal crops (for example, maize and barley) that are difficult to regenerate from protoplast culture.

It is generally agreed that decreasing the functional time of CRISPR/Cas9 is beneficial for lowering off-target effects of genome editing. Here, with RNP mediated genome editing, we found that off-target mutations were drastically reduced in the recipient wheat cells, and absent in the recovered mutant population (Figs 1b,c and 2c). Although reason for the very low level of off-target mutations in RNP mediated genome editing remains to be determined, we speculate that avoidance of CRISPR/Cas9 transgene integration, fast cellular degradation of the RNPs, and shortened functional time of CRISPR/Cas9 may be involved. In line with this proposition, the mutants generated using gw2-RNPs had much fewer mutated alleles and were more specific than those produced by pGE-TaGW2 (Fig. 2d). Therefore, genome editing using CRISPR/Cas9 RNPs may provide a general method for reducing unintended off-target cleavage.

The MPE we obtained for the alternative bread wheat variety YZ184 was lower than that for Kenong 199, which has been the model variety for our wheat genome editing research[12]. This indicates that the efficiency of CRISPR/Cas9 RNP based genome editing may be genotype dependent. We are now testing this possibility by working with more diverse bread wheat varieties. Nevertheless, an MPE of 1–2% may still be acceptable for most investigations considering the rapidity of our method (Fig. 3).

In summary, we established an efficient and specific CRISPR/Cas9 RNP mediated genome editing method for bread wheat. While our manuscript was under review, Svitashev et al.[13] also reported targeted mutagenesis in maize using the CRISPR/Cas9 ribonucleoprotein complexes. Our study, together with the preceding work on lettuce[8] and maize[13], suggest that the use of CRISPR/Cas9 RNPs will greatly aid specific and clean genome editing in a wide range of plant species. As the mutant plants edited using CRISPR/Cas9 RNPs do not have integrated transgenes, their application in practical breeding and commercialization should be more public acceptable, thus accelerating precision crop improvement.

## Methods

**Cas9 protein purification.** For *in vitro* expression of Cas9 protein, the Cas9 coding sequence flanked by two SV40 NLS signals was cloned into pET28a ( + ) to generate pET28a-Cas9-His. Cas9 protein was expressed in the *E. coli* Rosetta strain at 16 °C, purified by nickel affinity chromatography (Invitrogen, R90101), and dialyzed with Cas9 storage buffer (20 mM HEPES pH 7.5, 150 mM KCl, 1 mM DTT, and 3% glycerol). The purity and concentration of Cas9 protein were measured by SDS–polyacrylamide gel electrophoresis (SDS–PAGE) (Supplementary Fig. 5) and Bradford protein assay, respectively. The final concentration was adjusted to $\sim 2\,\mathrm{mg\,ml}^{-1}$.

**In vitro transcription of sgRNA.** The templates for transcription were amplified from pGE-TaGW2 or pGE-TaGASR7 using appropriate primers (Supplementary Table 3). Transcription was accomplished using the HiScribe T7 In Vitro Transcription Kit (New England Biolabs) according to the manufacturer's instructions.

**In vitro cleavage of CRISPR/Cas9 RNP.** DNA fragments containing the target site were amplified, purified and eluted with RNase-free water. Cas9 protein (1 μg) and sgRNA (1 μg) were mixed with the purified target DNA (100–150 ng) in Cas9 reaction buffer (20 mM HEPES, pH 7.5, 150 mM KCl, 10 mM MgCl₂, 0.5 mM DTT) in a total volume of 20 μl, followed by digestion at 37 °C for 0.5–1 h. The digested products were immediately separated on a 2% agarose gel, and cleavage activity was measured by the amount of digested products over the total amount of input target DNA. The DNA bands were quantified using Gel quantification software (Image J)[10].

**Protoplast assays.** Wheat protoplasts transformation was performed as previously described[14]. Briefly, wheat seeds were grown in nutrient-rich soil at 25 °C for two weeks with a photoperiod of 16 h light: 8 h dark. Cas9 protein (20 μg) and sgRNA (20 μg) were mixed and delivered into protoplasts by PEG-mediated transfection. After 48 h, the protoplasts were collected to extract DNA for PCR-RE assay.

**Biolistic bombardment of Cas9 RNP complexes.** The RNPs were delivered into the wheat immature embryos (cv Kenong 199) by particle bombardment according to Ortigosa et al.[15] with some modifications. For each shot, Cas9 protein (2 μg) and sgRNA (2 μg) were premixed in Cas9 reaction Buffer (20 mM HEPES, pH 7.5, 150 mM KCl, 10 mM MgCl₂, 0.5 mM DTT) in a total volume of 10 μl and incubated at 25 °C for 10 min. Then 5 μl gold nanoparticles (0.6 μm) were added

**Figure 3 | Flowchart of CRISPR/Cas9 RNP induced targeted mutagenesis in wheat.** RNP preparation takes 3-4 days (including Cas9 protein purification). RNP validation *in vitro* and in protoplasts takes 3–4 days (not including the time for plant preparation). RNP coating and delivery can be completed within 4 h. After this, the plantlets regenerate in 6–8 weeks. Identification of mutants by PCR-RE and DNA sequencing takes 3 days.

and the coated particles were spread directly onto the carrier and allowed to air-dry at room temperature for about 2 h. Biolistic bombardment and embryos processing have been described following our previous publication[3].

**Mutation identification in regenerated plants by PCR-RE assay.** To save cost and labour, we detected the mutations by pooling 3–4 seedlings derived from each immature embryo[3]. The positive mutants were further analysed with homoeolog specific primers and confirmed by Sanger sequencing.

**Deep amplicon sequencing.** Two days after bombardment with gw2-RNPs (or gasr7-RNPs) or pGE-TaGW2 DNA, immature embryos were collected to extract genomic DNA for deep sequencing. In the first round PCR, the target regions were amplified using site specific primers (Supplementary Table 3). In the second round PCR, forward and reverse barcodes were added to the end of the PCR product for library construction (Supplementary Table 3). Equal amount of PCR products were mixed as a pool and the samples were used for Illumina sequencing at Beijing Genome Institute. Indels, including insertions and deletions, occurring at the Cas9 cleavage sites were considered as mutations.

**Data availability.** The authors declare that all data supporting the findings of this study are available in the article and its Supplementary Information files or are available from the corresponding author on request.

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

## Acknowledgements

We thank Daowen Wang for critical reading of the manuscript and Jin-Long Qiu for helpful discussion and insightful comments on this manuscript. We also thank Haopeng Yu for deep sequencing analysis, Kangquan Yin for providing the pET28a-Cas9-His vector, and Huaigang Zhang for providing the immature wheat embryos of YZ814. This work was supported by the National Key Research and Development Program of China (2016YFD0101804), the Chinese Academy of Sciences (KFZD-SW-107), the Ministry of Agriculture of China (2014ZX0801003B) and the National Natural Science Foundation of China (31420103912, 31271795 and 31570369).

## Author contributions

Z.L. and C.G. designed the experiments; Z.L., T.L. and Y.Z. performed most of the experiments; Y.W., Q.Z. and C.L. purified Cas9 protein; J.L., H.Z. and Y.R. performed wheat tissue culture work; Z.L., K.C. and C.G. wrote the manuscript and C.G. supervised the project.

## Additional information

**Competing financial interests:** C.G., Z.L. and Y.W. filed a PCT patent application (Application Number PCT/CN2016/076244) based on the results reported in this paper. All other authors declare no competing financial interests.

**Publisher's note**: 

