## [Peer Review File · Nature Communications]

REVIEWERS' COMMENTS:

Reviewer #1 (Remarks to the Author):

The work by Liang et. al. entitled "Efficient DNA free genome editing of bread wheat using CRISPR/Cas9 ribonucleoprotein complex" attempted to improve the engineering of the wheat genome using the RNAP complex. The authors argue that the use of RNP complex would reduce the off-target effects and limit the possibilities of having DNA integrations from the input DNA of the CRISPR/Cas9 machinery. First they selected one genomic target that they used before in their previous research, and tested the efficacy of the complexed RNP against the genomic target in the protoplast. They demonstrated high activities of the RNP in the mutagenesis. Subsequently, they used this RNP complex to bombard immature embryos, and they successfully recovered mutant alleles albeit at much lower frequency, but this is expected. Additionally, they have demonstrated the improvements of the precision and reduction of the off-target activities by analyzing the off-targeting of one allele with one mismatch in the target sequence. There was a reduction in the off-target effects. Moreover, they used one more gRNA to target a different genomic target and a different genotype. They observed similar frequencies albeit at reduced levels and they argue that this is because the genotype dependent nature of the process encompassing both transformation/ regeneration and the editing with specific RNP.

Overall, this work advances the targeted engineering of cereal genomes and report clear advancements over the current methods. But the system and the whole process lack the novelty. Certainly, the work is important and with good impact given the importance of wheat for food security. But what is really new? I doubt that combination of these reagents to edit a couple of targets with established protocols would be substantial and novel. I expected also that more targets will be used to provide more quantitative and convincing data on the practicality, reproducibility of this method. I suggest that this finding may be reported in a different format like letter to editor or something similar.

Reviewer #2 (Remarks to the Author):

The MS by Liang et al. entitled "Efficient DNA-free genome editing of bread wheat using CRISPR/Cas9 2 ribonucleoprotein complex" describes an important advance in genome editing of a major staple crop, such as wheat. The authors report on application of Cas9/sgRNA ribonucleoprotein for editing wheat genome in a quick (7-9 weeks) and efficient way (mutagenesis efficiency above 4%). The resulting mutant plants are transgene-free. The latter factor is very important for acceptance of genetically edited crops as non-GM. I strongly recommend this MS for publication in Nature Communications.

Minor point:

The authors should cite the following paper:

Nat Commun. 2016 Nov 16;7:13274. doi: 10.1038/ncomms13274.

Genome editing in maize directed by CRISPR-Cas9 ribonucleoprotein complexes.
Svitashev S, Schwartz C, Lenderts B, Young JK, Mark Cigan A.

Reviewer #3 (Remarks to the Author):

This manuscript describes the use of CRISPR/Cas9 ribonucleoprotein (RNP) complexes to achieve gene editing in wheat. The key claims of the paper are that it is possible to obtain efficient editing in wheat using this 'DNA-free' method. The fact that it does not involve any DNA insertion is thought to offer advantages in terms of possible ease of commercialization. Secondly the paper claims that the RNP method leads to a significant reduction in off-target mutations.

This work follows on from previous work by the authors demonstrating CRISPR/Cas9 mediated gene editing in wheat by transient expression of either DNA or RNA. This previous method was also shown to produce edited plants that did not contain any transgenes but could not avoid possible nucleic acid insertion. The current paper describes the further development of using RNPs rather than DNA or RNA and thus removes the chance of nucleic acid insertion.

The use of RNPs is not novel as it has been previously reported as described by the authors, but the use in wheat, one of the most difficult crops to work with and one of the most important is novel.

Because of the potential impact of this technology in wheat I recommend publication. The data demonstrating the lack of off-target effects using this technique is compelling and is another reason for my recommendation. The efficiency of mutant production (28 mutants from 640 immature embryos - 4.4%) is sufficient to make this method widely applicable. In addition the authors were able to demonstrate use of the method in 2 different wheat genotypes.

My only other comment is that there are a few places where the English could be improved but this is very minor.

27 November 2016

REVIEWERS' COMMENTS:

Reviewer #1 (Remarks to the Author):

The work by Liang et. al. entitled "Efficient DNA free genome editing of bread wheat using CRISPR/Cas9 ribonucleoprotein complex" attempted to improve the engineering of the wheat genome using the RNAP complex. The authors argue that the use of RNP complex would reduce the off-target effects and limit the possibilities of having DNA integrations from the input DNA of the CRISPR/Cas9 machinery. First they selected one genomic target that they used before in their previous research, and tested the efficacy of the complexed RNP against the genomic target in the protoplast. They demonstrated high activities of the RNP in the mutagenesis. Subsequently, they used this RNP complex to bombard immature embryos, and they successfully recovered mutant alleles albeit at much lower frequency, but this is expected. Additionally, they have demonstrated the improvements of the precision and reduction of the off-target activities by analyzing the off-targeting of one allele with one mismatch in the target sequence. There was a reduction in the off-target effects. Moreover, they used one more gRNA to target a different genomic target and a different genotype. They observed similar frequencies albeit at reduced levels and they argue that this is because the genotype dependent nature of the process encompassing both transformation/regeneration and the editing with specific RNP.

Overall, this work advances the targeted engineering of cereal genomes and report clear advancements over the current methods. But the system and the whole process lack the novelty. Certainly, the work is important and with good impact given the importance of wheat for food security. But what is really new? I doubt that combination of these reagents to edit a couple of targets with established protocols would be substantial and novel. I expected also that more targets will be used to provide more quantitative and convincing data on the practicality, reproducibility of this method. I suggest that this finding may be reported in a different format like letter to editor or something similar.

Response: Thank you for the comments, which are very constructive and helpful. Your support of our work is also highly appreciated. Since the submission of our work to NC, we have edited more bread wheat genes in wheat protoplasts, and found that our RNP-based genome editing tool is robust. The timely publication of our work in NC will enable more wheat researchers to use the reported method as quickly as possible, thus helping international efforts in wheat crop improvement.

Reviewer #2 (Remarks to the Author):

The MS by Liang et al. entitled "Efficient DNA-free genome editing of bread wheat using CRISPR/Cas9 2 ribonucleoprotein complex" describes an important advance in genome editing of a major staple crop, such as wheat. The authors report on application of Cas9/sgRNA ribonucleoprotein for editing wheat genome in a quick (7-9 weeks) and

efficient way (mutagenesis efficiency above 4%). The resulting mutant plants are transgene-free. The latter factor is very important for acceptance of genetically edited crops as non-GM.

I strongly recommend this MS for publication in Nature Communications.

Minor point:

The authors should cite the following paper:

Nat Commun. 2016 Nov 16;7:13274. doi: 10.1038/ncomms13274.

Genome editing in maize directed by CRISPR-Cas9 ribonucleoprotein complexes.

Svitashev S, Schwartz C, Lenderts B, Young JK, Mark Cigan A.

Response: Thank you for the overall positive assessment. We have added the reference you suggested to our revised manuscript.

Reviewer #3 (Remarks to the Author):

This manuscript describes the use of CRISPR/Cas9 ribonucleoprotein (RNP) complexes to achieve gene editing in wheat. The key claims of the paper are that it is possible to obtain efficient editing in wheat using this 'DNA-free' method. The fact that it does not involve any DNA insertion is thought to offer advantages in terms of possible ease of commercialization. Secondly the paper claims that the RNP method leads to a significant

reduction in off-target mutations.

This work follows on from previous work by the authors demonstrating CRISPR/Cas9 mediated gene editing in wheat by transient expression of either DNA or RNA. This previous method was also shown to produce edited plants that did not contain any transgenes but could not avoid possible nucleic acid insertion. The current paper describes the further development of using RNPs rather than DNA or RNA and thus removes the chance of nucleic acid insertion.

The use of RNPs is not novel as it has been previously reported as described by the authors, but the use in wheat, one of the most difficult crops to work with and one of the most important is novel.

Because of the potential impact of this technology in wheat I recommend publication. The data demonstrating the lack of off-target effects using this technique is compelling and is another reason for my recommendation. The efficiency of mutant production (28 mutants from 640 immature embryos - 4.4%) is sufficient to make this method widely applicable. In addition the authors were able to demonstrate use of the method in 2 different wheat genotypes.

My only other comment is that there are a few places where the English could be improved but this is very minor.

Response: Thank you for your kind support. The revised manuscript has been reviewed and edited by an English biologist.